# Disruption of *yqhG* attenuates virulence in *methicillin-resistant Staphylococcus aureus* by compromising membrane stability and oxidative stress resistance

Jianhua Liao, Jun Cheng, Baoqing Liu, Yuzhi Shao, Chunyan Meng ⬤*

Department of Surgery, Zhejiang Hospital, Zhejiang, China

* mcyzjyy@163.com

## Abstract

The growing prevalence of methicillin-resistant *Staphylococcus aureus* (MRSA) infections, coupled with the increasing resistance to existing antibiotics, underscores the critical need for novel therapeutic approaches to combat this pathogen. In this study, the role of *yqhG*, a conserved gene encoding a periplasmic protein, in MRSA virulence and stress adaptation was investigated. *yqhG* deletion in MRSA significantly attenuated virulence in a murine infection model, leading to reduced bacterial burden in infected organs and improved host survival. In vitro, the *yqhG* mutant exhibited impaired membrane integrity, reduced motility, and increased sensitivity to oxidative stress, but did not affect biofilm formation. These defects were fully restored upon genetic complementation. These findings highlight the critical role of *yqhG* in maintaining MRSA's ability to withstand host-imposed stresses, suggesting that *yqhG* is a key determinant of MRSA pathogenesis. The study provides new insights into the stress-defense mechanisms employed by MRSA and underscores *yqhG* as a potential target for therapeutic strategies aimed at combating MRSA infections.

## Introduction

*Staphylococcus aureus* is a ubiquitous commensal and opportunistic pathogen in humans [1]. While it often resides harmlessly on skin and mucosal surfaces, *S. aureus* can cause a spectrum of diseases—from superficial abscesses to life-threatening endocarditis, septicemia, and pneumonia [1–3]. Its success reflects a large arsenal of virulence factors (toxins, adhesins, immune-evasion proteins) that promote tissue colonization and immune escape [4]. Alarmingly, *S. aureus* strains have repeatedly acquired antibiotic resistances [5–7]. In particular, methicillin-resistant *S. aureus* (MRSA) – defined by acquisition of the *mecA* gene encoding PBP2a – has emerged as a leading cause of healthcare- and community-associated infections [4]. MRSA poses a major public health challenge: for example, the U.S.

**Data availability statement:** All relevant data are within the manuscript and its supplementary materials.

**Funding:** This study was partly supported by a grant from the Zhejiang Traditional Chinese Medicine Administration (Grant No. 2023ZL221, awarded to CM). The funder's website is: http://www.zjwjw.gov.cn/. The funder had no role in study design, data collection and analysis, decision to publish, or preparation of the manuscript.

**Competing interests:** The authors have declared that no competing interests exist.

CDC classifies MRSA as a "Serious Threat," with on the order of $10^5$–$10^6$ infections and ~$10^4$ deaths annually [8]. Thus, understanding the factors that govern MRSA's ability to adapt to host environments and cause disease is of critical importance.

Bacterial pathogens routinely encounter hostile conditions during infection, and the ability to sense and respond to stress is crucial for survival and virulence [9,10]. Indeed, stress response pathways often enhance pathogenicity: bacteria that mount robust defenses against environmental insults tend to exhibit greater fitness, antibiotic tolerance, biofilm formation, and immune evasion [11]. A classic example is oxidative stress: during infection host phagocytes unleash a burst of reactive oxygen species (ROS) and chlorine species (e.g., hydrogen peroxide and HOCl) as microbicidal weapons [1]. To counter this, *S. aureus* encodes powerful antioxidant systems (catalase, superoxide dismutases, carotenoid pigments, thiol-based redox proteins) to detoxify ROS and repair damage. At the same time, preserving cell envelope integrity under stress is equally vital. For instance, the *S. aureus msaABCR* operon is induced by oxidative challenge and reinforces the peptidoglycan cell wall during $H_2O_2$ exposure [1]. By maintaining membrane integrity and cell viability, such stress-response systems enable MRSA to persist in hostile host niches. In sum, pathways that protect against oxidative or envelope stress are intimately linked to bacterial adaptation and virulence.

Despite the importance of stress defenses, many potential *S. aureus* stress-response factors remain uncharacterized. In other bacteria, however, related proteins have been implicated in stress tolerance and pathogenicity. Bessaiah *et al.* (2019) found that deletion of *yqhG* in a clinical *E. coli* strain caused a profound virulence defect: fimbrial adhesin (type 1 fimbriae) expression and bladder colonization were markedly reduced, while the mutant became hyper-motile and strikingly sensitive to hydrogen peroxide [12]. In other words, loss of YqhG impaired redox homeostasis and stress resistance and correspondingly diminished pathogenic fitness. These findings suggest that YqhG contributes to detoxification pathways or oxidative stress defenses in Gram-negative pathogens. (Other Yqh-family proteins, such as the YqhD NADPH-dependent aldehyde reductase, are known to detoxify reactive carbonyls [13], highlighting the potential for this family in redox metabolism.) Importantly, no analogous role of *yqhG* has been described in *S. aureus* or MRSA.

In this study, the *yqhG* gene in MRSA was characterized to fill this knowledge gap. The effects of *yqhG* deletion on key pathogenic traits—including membrane integrity, motility, and survival under oxidative stress—were assessed. By elucidating the contribution of *yqhG* to MRSA pathogenesis and stress adaptation, this work aimed to uncover novel aspects of MRSA virulence and identify potential targets for intervention.

## Materials & Methods

### Bacterial strains and culture conditions

Detailed information on all bacterial strains and plasmids used in this study is provided in Table 1. Methicillin-resistant *Staphylococcus aureus* (MRSA) USA300-LAC was used as the wild-type (WT) background strain [14]. The *yqhG* deletion mutant (Δ*yqhG*) was constructed via allelic exchange using the temperature-sensitive vector pKOR1 [15,16].

**Table 1. Bacterial Strains and Plasmids Used in This Study.**

| Strain/Plasmid | Genotype/ Description | Antibiotic Resistance | Reference |
|---|---|---|---|
| USA300-LAC (WT) | Wild-type MRSA strain | Erythromycin | This study |
| ΔyqhG | yqhG deletion mutant | None | This study |
| ΔyqhG-C | ΔyqhG carrying pYJ335-yqhG | Erythromycin | This study |
| RN4220 | Intermediate strain used for plasmid transfer | Erythromycin | ATCC |
| pKOR1 | Temperature-sensitive allelic exchange vector | Kan$^r$ | Bae et al., 2004 |
| pYJ335 | Staphylococcal shuttle vector; Cm$^r$ | Cm$^r$ | This study |
| pYJ335- yqhG | Complementation vector carrying yqhG with native promoter | Cm$^r$ | This study |

Briefly, approximately 1 kb upstream and downstream regions flanking the *yqhG* coding sequence were amplified by PCR and ligated by overlap extension PCR. All oligonucleotide primers used for gene deletion and complementation are listed in Table 2. The deletion construct was cloned into pKOR1 and introduced into USA300 via RN4220 intermediate strain. Allelic exchange was induced via temperature shift (43 °C) and counterselection on anhydrotetracycline (1 µg/mL). Mutants were confirmed by colony PCR and Sanger sequencing. For genetic complementation, the *yqhG* open reading frame along with its native promoter (~300 bp upstream) was cloned into the shuttle vector pYJ335. The resulting plasmid (pYJ335-*yqhG*) and empty vector were electroporated into the Δ*yqhG* mutant. All strains were routinely cultured in tryptic soy broth (TSB; Oxoid) or on TSB agar at 37 °C. For plasmid maintenance, erythromycin (10 µg/mL) or chloramphenicol (10 µg/mL) was added where appropriate.

### In vitro growth curve analysis

Overnight cultures of WT, Δ*yqhG*, and Δ*yqhG*-C were diluted 1:100 into fresh TSB and grown at 37 °C with shaking (250 rpm). Optical density at 600 nm ($OD_{600}$) was recorded every 2–3 hours over an 18-hour period using a spectrophotometer (BioPhotometer, Eppendorf). Three biological replicates were measured per strain.

### Animals and Housing

Female BALB/c mice (6–8 weeks old, 18–20 g) were obtained and maintained under specific pathogen-free (SPF) conditions. Mice were housed 5 per cage in a pathogen-free animal facility with a 12:12-hour light–dark cycle, constant temperature (~22°C), and humidity control. Soft bedding and nesting material were provided for environmental enrichment, and handling was minimized to reduce stress. All animal procedures were conducted in compliance with the *NIH Guide for the Care and Use of Laboratory Animals* (NIH Publication No. 8023, revised 1978) and approved by the Institutional Animal Care and Use Committee (IACUC) of [Institute Name] (Protocol #TIRM-IACUC-2024–0532). Notably, the experimental design anticipated some mortality due to the infection; death or moribundity as an endpoint was explicitly approved by the IACUC in the protocol.

### Infection Procedure

A systemic MRSA infection was induced via tail vein injection. Each mouse was inoculated with $2 \times 10^7$ colony-forming units (CFU) of methicillin-resistant *Staphylococcus aureus* (MRSA) in 100 µL of sterile phosphate-buffered saline (PBS). The inoculation dose was selected based on preliminary dose–response experiments and previously published MRSA infection models, which indicated that this bacterial load reliably induces a reproducible systemic infection with measurable bacterial dissemination while maintaining an acceptable survival window for comparative analysis [17]. Injections were performed using a 27-gauge needle over ~5 seconds to ensure proper delivery into the circulation. Mice were observed for a few minutes post-injection to confirm recovery from the brief restraint and to monitor for any acute adverse reactions.

**Table 2. Primers Used in This Study.**

| Primer Name | Sequence (5'-3') | Purpose |
|---|---|---|
| *yqhG*_up-F | ATGCGTCTAGAACGTTTCAGCAGGTA | Amplify upstream region for allelic exchange |
| *yqhG*_up-R | TCAGTTGTTTCTCGGCTAGTTG | Amplify upstream region for allelic exchange |
| *yqhG*_down-F | GAACTAGCCGAGAAACAACCTGA | Amplify downstream region for allelic exchange |
| *yqhG*_down-R | CGTCAAGCTTTTAGGCCAGGTCAT | Amplify downstream region for allelic exchange |
| *yqhG*-comp-F | GATCGAATTCATGAGTGAAGTCGACAG | Amplify *yqhG* with promoter |
| *yqhG*-comp-R | TCGACTCTAGATTAGTGATGTGGTTTG | Amplify *yqhG* with promoter |

## Survival Study

For the survival analysis, n = 10 mice per group were infected as described above and monitored for survival over a 5-day period. Health and behavior were assessed at least twice daily (approximately every 12 hours) throughout the study. During each observation, mice were checked for general appearance, activity, and clinical signs of illness or distress. Survival was recorded daily, and any mouse that met predefined humane endpoint criteria (see *Monitoring and Humane Endpoints* below) was humanely euthanized to prevent undue suffering. The 5-day observation period was chosen based on preliminary studies indicating that most mortality in this model occurs within this timeframe.

## Bacterial Burden Study

For the bacterial burden analysis, n = 5 mice per group were infected identically and euthanized at 24 hours post-infection. At the 24 h time point, mice were humanely sacrificed (as described under *Monitoring and Humane Endpoints*) to collect tissues for bacterial load quantification. Organs such as the kidneys, spleen, and blood were aseptically harvested for CFU determination on selective media. This 24-hour endpoint was selected to evaluate early infection burden before significant mortality occurred. All animals in this cohort were euthanized at the predetermined time point regardless of clinical status, using the same approved method of euthanasia.

## Monitoring and Humane Endpoints

Mice were closely monitored for signs of morbidity throughout the experiment, and predefined humane endpoints were applied to minimize suffering. Clinical signs that warranted euthanasia (humane endpoints) included, but were not limited to:

- Weight loss >20% of baseline body weight (or rapid weight loss over a short period).

- Hypothermia, indicated by body temperature falling significantly below normal (subnormal).

- Severe lethargy or hunched posture, such as a persistent hunch, inactivity, or prolonged recumbency.

- Unresponsiveness to touch or failure to respond to external stimuli (near-moribund state).

- Labored breathing (dyspnea or gasping).

Mice exhibiting any one or a combination of these signs were considered to have reached a humane endpoint. Such animals were euthanized within 1–2 hours of observing the criteria to prevent further distress. Euthanasia was performed by $CO_2$ inhalation in a chamber, followed by cervical dislocation to ensure death, in accordance with the AVMA Guidelines for the Euthanasia of Animals (2020), which recommend the use of a secondary physical method for rodent euthanasia to guarantee a humane and effective death. All euthanasia procedures were carried out by trained personnel in a manner designed to minimize fear or discomfort (e.g., using the home cage as the euthanasia chamber when practical, to avoid

stress from transfer). If any animal was found dead in the cage (i.e., an unanticipated death before meeting euthanasia criteria or before the planned endpoint), the event was immediately documented and reported to the IACUC as required.

## Ethical Considerations

No post-infection analgesics or anesthetics were administered to the mice in these studies. This decision was made to avoid potential interference with disease progression and immune responses, as some analgesic drugs can modulate infection outcomes. The omission of pain relief was carefully reviewed and approved by the IACUC in light of the scientific necessity, and it was counterbalanced by the use of stringent humane endpoints and frequent monitoring as described above.

All research staff involved in animal care and procedures were trained and certified in proper animal handling, clinical monitoring, and euthanasia techniques. This ensured that signs of pain or distress were recognized promptly and that all interventions (injections, observations, and euthanasia) were performed competently and humanely. Throughout the study, animal welfare was prioritized: mice were maintained in a clean, enriched environment with minimal handling to reduce stress, and their health status was checked at least every 12 hours (or more often if any signs of illness were noted). All methods and endpoints described in this section adhered to the AVMA 2020 euthanasia guidelines and the NIH animal care guidelines, ensuring that the experiments were conducted with the highest ethical standards for animal research.

## Biofilm assay

Biofilm formation was assessed using a standard microtiter dish crystal violet assay [18]. Overnight cultures of each strain were diluted 1:100 in TSB supplemented with 1% glucose. 200 μL aliquots were dispensed into 96-well flat-bottom polystyrene plates (Costar) and incubated statically at 37 °C for 24 h. Wells were gently washed twice with PBS, air-dried, stained with 0.1% crystal violet for 15 min, and washed again. The dye was solubilized in 200 μL of 33% acetic acid, and absorbance was measured at 570 nm using a microplate reader (BioTek Synergy HT).

## SYTOX Green membrane permeability assay

To assess membrane integrity, bacterial cultures were grown to $OD_{600} \approx 0.4$, harvested, and washed with PBS. Cell suspensions were adjusted to $OD_{600} = 0.2$ and incubated with 1 μM SYTOX Green (Thermo Fisher Scientific) for 30 min at room temperature in the dark. Fluorescence intensity was measured using a microplate reader (excitation 488 nm, emission 523 nm). Results were expressed as relative fluorescence units (RFU). All assays were performed in triplicate.

## Motility assay

Motility was assessed using a soft agar colony spreading assay [19,20]. TSB plates containing 0.24% agar were poured and dried for 30 min. Two microliters of overnight cultures were spotted onto the center of the plate and incubated at 37 °C for 12 h. Colony diameters were measured in millimeters along two perpendicular axes, and the average was recorded. Three independent experiments were performed with triplicate plates per strain.

## Measurement of cell-associated PSMs

Cell-associated phenol-soluble modulins (PSMs) were quantified following a modified extraction protocol optimized for surface-bound peptides [21]. Briefly, *Staphylococcus aureus* cultures were initiated by inoculating 50 μL of overnight culture into 5 mL of fresh TSB and incubated aerobically at 37 °C for approximately 18–20 h with shaking. Bacterial cells were collected by centrifugation at 2,500 × g for 20 min and washed once with PBS to remove residual medium. The resulting pellet was resuspended in 300 μL of 6 M guanidine hydrochloride and vortexed vigorously for 10 min to extract PSMs loosely associated with the cell surface. After centrifugation at 20,000 × g for 5 min, the supernatant was transferred

to a clean tube and evaporated to dryness under vacuum. The dried extract was reconstituted in 1 mL of 40% (v/v) acetonitrile containing 0.1% trifluoroacetic acid (TFA), vortexed for 10 min, and centrifuged again under the same conditions to remove insoluble debris. Approximately 0.8 mL of the clarified solution was dried once more and finally dissolved in 300 μL of ultrapure water. The samples were analyzed by reversed-phase HPLC equipped with a C4 analytical column (4.6 × 150 mm, 5 μm particle size) using a gradient of acetonitrile in 0.1% TFA at a flow rate of 1 mL min$^{-1}$, with detection at 215 nm. Individual PSM peaks were verified by comparison with synthetic standards and reference strains producing defined PSM species. The amount of each PSM was calculated from calibration curves constructed with known concentrations of synthetic peptides, and data were normalized to culture density ($OD_{600}$) or CFU counts to allow cross-sample comparison. All extractions and analyses were performed in triplicate to ensure reproducibility.

### Hydrogen peroxide sensitivity assay

The sensitivity of strains to oxidative stress was determined by measuring survival after hydrogen peroxide ($H_2O_2$) challenge [22]. Overnight cultures were washed and diluted to $1 \times 10^8$ CFU/mL in PBS. Each strain was exposed to 1.5% (v/v) hydrogen peroxide ($H_2O_2$, Sigma) at 37 °C for 20 min. At 0 and 20 min, samples were serially diluted, plated on TSB agar, and incubated overnight at 37 °C. Viability was calculated as $log_{10}$ reduction in CFU compared to the initial time point. Control groups without $H_2O_2$ were included to confirm viability. Each experiment was performed in triplicate.

### Statistical analysis

All quantitative data are presented as mean ± s.e.m. from three independent experiments. Statistical analysis was performed using one-way ANOVA for quantitative comparisons and the log-rank test for survival analysis. $P < 0.05$ was considered statistically significant. Analyses were performed using GraphPad Prism (version 9.0).

## Results

### YqhG is required for systemic virulence but dispensable for *in vitro* growth

To evaluate whether *yqhG* influences basic bacterial physiology, the growth dynamics of MRSA USA300-LAC wild-type (WT), Δ*yqhG*, and the complemented strain (Δ*yqhG*-C) were monitored in tryptic soy broth (TSB) under aerobic conditions at 37 °C. All three strains displayed nearly identical growth curves, suggesting that *yqhG* is not essential for proliferation under nutrient-rich, laboratory conditions (Fig 1A, S1 Table). These data confirm that deletion of *yqhG* does not impair fitness *in vitro*. In contrast, *in vivo* experiments revealed a profound impact of *yqhG* on virulence. BALB/c mice were challenged via tail vein injection with $2 \times 10^7$ CFU of WT, Δ*yqhG*, or Δ*yqhG*-C strains and monitored for survival over five days. Mice infected with the Δ*yqhG* mutant showed markedly improved survival relative to those challenged with the WT or complemented strain (Fig 1B, S2 Table), indicating attenuated virulence. To further assess bacterial dissemination during infection, multiple organs were analyzed 24 h post-infection. Quantification of colony-forming units (CFU) from homogenized tissues revealed that Δ*yqhG*-infected mice had substantially reduced bacterial burdens in the heart, kidneys, liver, and spleen, whereas the complemented strain restored colonization to near wild-type levels (Fig 1C–F, Tables S3–S6). These data provide comprehensive evidence that *yqhG* is dispensable for growth under laboratory conditions but indispensable for full virulence and systemic dissemination during MRSA infection.

### Biofilm formation is not dependent on YqhG

To assess whether *yqhG* contributes to surface-associated community development, biofilm formation was quantified using a static crystal violet staining assay in TSB supplemented with 1% glucose. WT, Δ*yqhG*, and complemented strains were grown for 24 hours, after which adherent biomass was stained and measured spectrophotometrically. All strains exhibited comparable levels of biofilm biomass (Fig 2, S7 Table), and no statistically significant differences

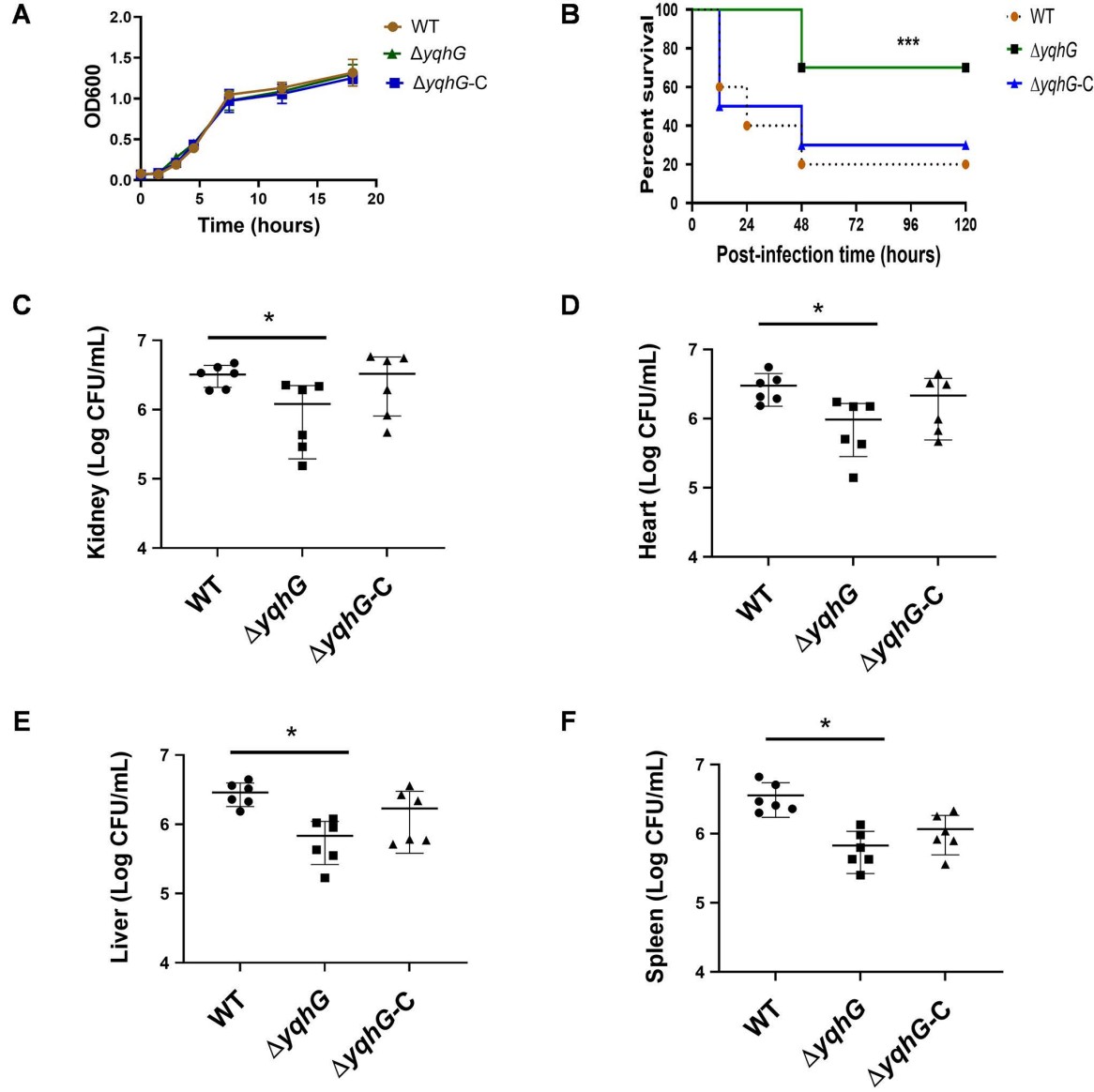

**Fig 1. YqhG is required for systemic virulence but not for *in vitro* growth.** (A) Growth curves of MRSA wild-type (WT), Δ*yqhG* mutant, and complemented strain (Δ*yqhG*-C) cultured in TSB at 37 °C with shaking. (B) Kaplan–Meier survival analysis of BALB/c mice (n = 10 per group) following intravenous infection with 2 × 10⁷ CFU of each strain. (C-F) Bacterial burden in the heart (C), kidney (D), liver (E) and spleen (F) at 24 h post-infection. CFU were enumerated from homogenized tissues. Data represent at least three independent experiments. Statistical analyses were performed using one-way ANOVA (C-F) and log-rank test (B). *$P < 0.05$, **$P < 0.01$, ***$P < 0.001$.

were observed. These results indicate that *yqhG* does not play a detectable role in biofilm formation under the tested conditions.

## YqhG is critical for maintaining membrane integrity

Because many virulence factors influence bacterial envelope structure, membrane integrity was examined using SYTOX Green, a nucleic acid-binding dye that penetrates only damaged membranes. Following a 30-minute incubation with the

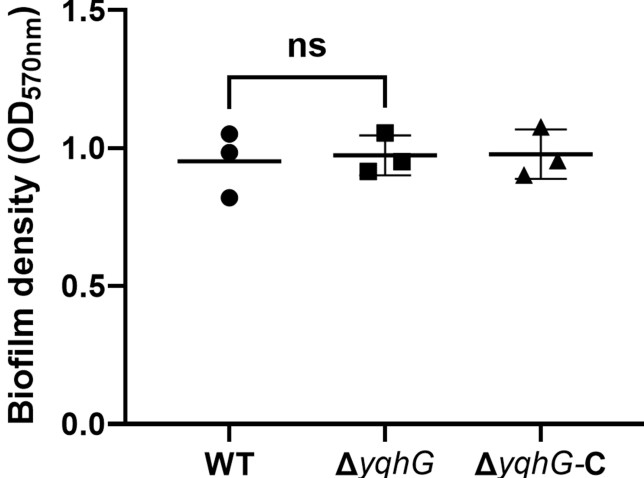

**Fig 2. Biofilm formation in MRSA strains.** Quantification of biofilm formation by wild-type (WT), Δ*yqhG*, and complemented (Δ*yqhG*-C) strains using crystal violet staining after static incubation in TSB + 1% glucose. Absorbance at 570 nm reflects biofilm biomass. Bars represent the mean ± s.e.m. of three independent experiments. Statistical analysis was performed using one-way ANOVA. ns, not significant.

dye, the Δ*yqhG* strain showed elevated fluorescence intensity compared to WT, consistent with increased membrane permeability (Fig. 3, Table S8). Restoration of *yqhG* expression in the complemented strain reduced SYTOX Green uptake to levels comparable with the WT strain, suggesting that the loss of *yqhG* compromises membrane stability. This defect may contribute to the attenuation of virulence observed in the infection model.

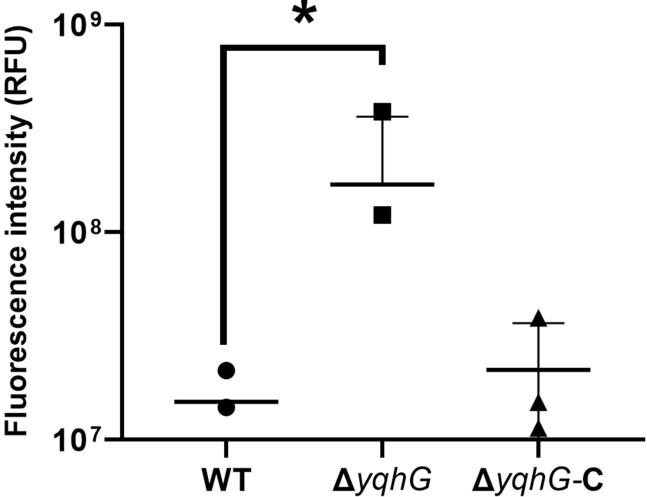

**Fig 3. Membrane integrity in the *yqhG* mutant.** Fluorescence-based quantification of membrane permeability in wild-type (WT), Δ*yqhG*, and complemented (Δ*yqhG*-C) strains using SYTOX Green uptake. Bacteria were incubated with 1 μM SYTOX Green, and fluorescence intensity (RFU) was measured after 30 min. Bars represent mean ± s.e.m. from three independent experiments. Statistical analysis was performed using one-way ANOVA. *$P < 0.05$.

## YqhG facilitates surface-associated motility

To explore whether *yqhG* affects bacterial motility, colony spreading assays were performed on low-agar (0.24%) plates. After 12 hours of incubation at 37 °C, WT and Δ*yqhG*-C strains exhibited typical outward radial expansion, whereas Δ*yqhG* colonies remained more compact with a substantially smaller diameter (Fig. 4, S9 Table). This phenotype suggests impaired surface translocation, potentially reflecting changes in membrane fluidity, surfactant production, or flagella-independent motility mechanisms. Complementation with *yqhG* fully rescued the spreading defect, supporting a functional role for YqhG in facilitating motility.

## YqhG facilitates cell surface phenol-soluble modulins

To determine whether the reduced surface motility observed in the Δ*yqhG* mutant was associated with altered production of cell surface phenol-soluble modulins (PSMs), the total amount of cell-associated PSMs was quantified using guanidine hydrochloride extraction followed by reversed-phase HPLC analysis. The Δ*yqhG* strain exhibited a markedly lower abundance of surface-bound PSMs compared with the wild-type and complemented strains (Fig 5, S10 Table). Restoration of *yqhG* expression in the complemented strain reinstated PSM levels to those of the wild type, confirming that the defect was specifically attributable to *yqhG* disruption.

## YqhG promotes oxidative stress resistance

Given the central importance of oxidative stress resistance in host-pathogen interactions, the ability of each strain to tolerate hydrogen peroxide was examined. Cultures of WT, Δ*yqhG*, and Δ*yqhG*-C strains were exposed to 1.5% $H_2O_2$ for 20 minutes at 37 °C. Surviving bacteria were quantified by CFU enumeration. The Δ*yqhG* mutant exhibited a substantial reduction in viability following peroxide exposure, indicative of hypersensitivity to oxidative stress (Fig. 6, Table S11). Complemented strains displayed restored resistance, further confirming that YqhG contributes to the oxidative stress defense network of *S. aureus*. This phenotype may partly explain the reduced persistence of the Δ*yqhG* strain in murine tissues.

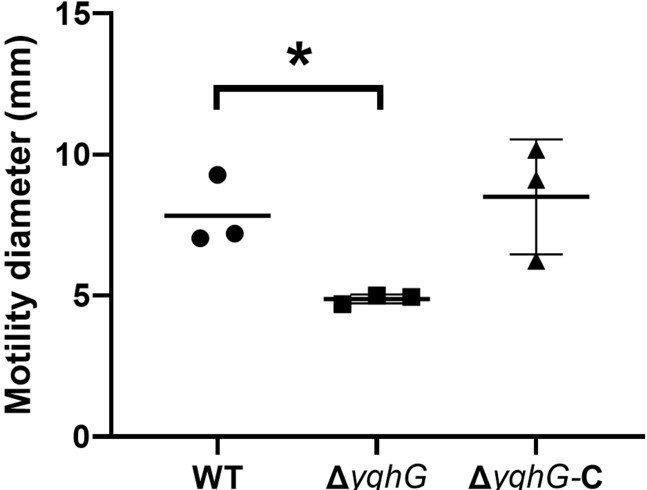

**Fig 4. Surface motility of the *yqhG* mutant.** Quantification of colony spreading on soft agar plates by wild-type (WT), Δ*yqhG*, and complemented (Δ*yqhG*-C) strains. Bacteria were spotted onto 0.24% agar and incubated at 37 °C for 12 h. Motility was assessed by measuring colony diameter (mm). Bars represent mean±s.e.m. of three independent experiments. Statistical analysis was performed using one-way ANOVA. * P<0.05.

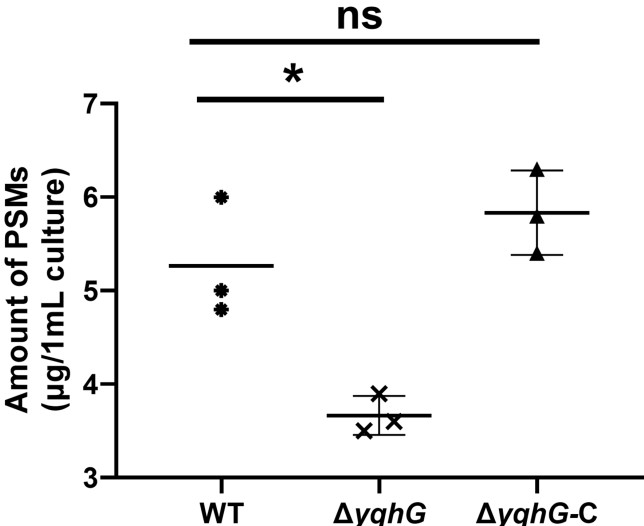

**Fig 5. Quantification of cell-associated PSMs in wild-type, ΔyqhG, and complemented strains.** Cell-associated PSMs were extracted from bacterial pellets using 6 M guanidine hydrochloride and quantified by reversed-phase HPLC. The ΔyqhG mutant exhibited a significant reduction in surface-bound PSMs compared with the wild-type strain, whereas complementation restored PSM levels to near wild-type values. Data represent mean±s.e.m. from three independent experiments. Statistical analysis was performed using one-way ANOVA; *$P<0.05$, ns, not significant.

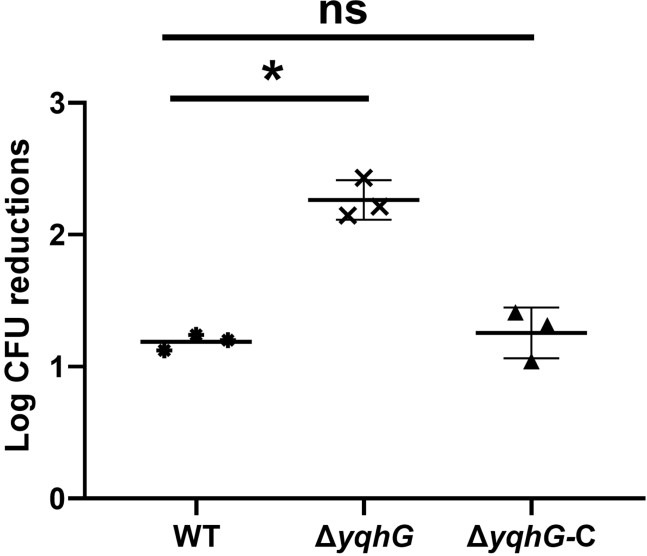

**Fig 6. Oxidative stress resistance of the yqhG mutant.** $Log_{10}$ CFU reductions in wild-type (WT), ΔyqhG, and complemented (ΔyqhG-C) strains after 20 min exposure to 1.5% hydrogen peroxide ($H_2O_2$) at 37 °C. Bacterial viability was assessed by serial dilution and CFU enumeration. Bars represent mean±s.e.m. of three independent experiments. Statistical analysis was performed using one-way ANOVA. $P<0.05$; ns, not significant.

## Discussion

Disruption of the *yqhG* gene produced a striking attenuation of MRSA virulence: the Δ*yqhG* mutant showed significantly reduced organ colonization and improved host survival *in vivo*. These phenotypes coincided with increased membrane permeability, impaired surface motility, and heightened sensitivity to oxidative stress. Notably, biofilm formation was unchanged by loss of *yqhG*, indicating that this factor specifically affects stress-resistance and envelope functions rather than biofilm-mediated persistence. Together, these data imply that YqhG supports MRSA pathogenesis by bolstering cell-envelope integrity and redox homeostasis under host-imposed stress.

The role of *yqhG* in MRSA mirrors key aspects of prior findings in other bacteria. In uropathogenic *E. coli*, Bessaiah et al. showed that *yqhG* encodes a predicted periplasmic lipoprotein required for full virulence [12]. Deletion of *yqhG* in *E. coli* CFT073 decreased expression of adhesive type-1 fimbriae and reduced bladder and kidney colonization, while increasing motility and sensitivity to hydrogen peroxide [12]. Similarly, the MRSA Δ*yqhG* mutant was hypersensitive to oxidative killing, suggesting a conserved role in redox defense. However, one notable difference is motility: *E. coli* Δ*yqhG* became more motile [12], whereas MRSA Δ*yqhG* showed impaired motility. This contrast may reflect fundamental differences in motility mechanisms; *S. aureus* lacks flagella and instead exhibits surfactant-driven sliding motility on soft agar (Agr/PSM-dependent) [23]. The marked decrease in motility observed in the MRSA Δ*yqhG* mutant implies that YqhG plays an important role in maintaining surface properties required for efficient colony spreading. In *S. aureus*, spreading motility is known to depend on amphipathic peptides such as PSMs, which act as natural surfactants that lower surface tension and enable passive translocation of bacterial clusters across moist surfaces [21]. Consistent with this, our quantitative analysis revealed that deletion of *yqhG* resulted in a significant reduction in cell-associated PSMs, while complementation restored their levels to those of the wild type. This finding suggests that YqhG may indirectly modulate surfactant availability at the cell envelope—either by influencing PSM synthesis, secretion, or surface retention. Given that YqhG is predicted to be a membrane-associated or periplasmic protein involved in maintaining envelope stability, its absence could alter membrane fluidity or charge distribution, thereby impairing the proper localization or accumulation of these amphipathic peptides. Reduced PSM levels would in turn compromise the ability of bacterial cells to spread efficiently on semi-solid surfaces.

YqhG homologs (often annotated YbjP/YqhG family) are widely conserved in bacteria, implying an ancient envelope-related function. The DUF3828 domain carried by YqhG belongs to the NTF2-like superfamily, which likely serves a non-catalytic ligand-binding role [24]. This structural insight suggests that YqhG might bind a small molecule or cell-wall component, thereby regulating activity of adjacent or operonic enzymes (e.g., peptidoglycan hydrolases or other envelope modulators) [24]. Consistent with this, the elevated membrane leakage in the Δ*yqhG* mutant implies a compromised cell-envelope. YqhG might therefore contribute to envelope biogenesis or remodeling – perhaps by delivering or sensing lipid or peptidoglycan precursors – analogous to how periplasmic lipoproteins participate in outer-membrane assembly in Gram-negatives. The accumulation of reactive oxygen species in host tissues would exacerbate any envelope defects, which could explain why loss of YqhG also heightens oxidative sensitivity and attenuates virulence. In short, *yqhG* may act at the nexus of membrane integrity and redox balance, helping MRSA to maintain envelope homeostasis during host attack.

These findings fit into a broader paradigm in which bacterial stress responses are intimately linked to pathogenesis. Host innate defenses generate oxidative stress and cell-envelope stress (e.g., cationic peptides, cell-wall-targeting enzymes), and pathogens must sense and counteract these insults to survive. *S. aureus*, for example, mounts robust defenses against oxidative burst by producing catalases, peroxidases and redox-balancing factors [1]. The fact that YqhG loss sensitizes MRSA to peroxide suggests it contributes to these defenses. Likewise, envelope-sensing regulatory systems in staphylococci – such as the intramembrane-sensing NsaRS two-component system – respond to cell-wall damage and are required for full virulence. NsaS-null mutants show altered cell-envelope architecture and are profoundly impaired in virulence-related traits (biofilm formation, survival in blood, resistance to phagocytes) [25]. The parallels imply that YqhG may function in concert with, or as part of, the cell-envelope stress response. By preserving membrane stability under stress, YqhG would help MRSA evade immune clearance and tolerate antimicrobial peptides. The absence of an

effect on biofilm formation in the ΔyqhG strain is also informative; it indicates that yqhG's contribution is specific to acute stress resistance rather than chronic sessile growth, distinguishing it from some global stress regulators.

In summary, yqhG emerges as a previously unrecognized determinant of MRSA pathogenesis, required for coping with host-derived stresses and maintaining envelope integrity. These findings expand our understanding of how bacteria integrate stress resistance with virulence, and suggest new strategies for disarming MRSA by targeting its stress-response.

## Conclusions

This study identifies yqhG as a crucial factor for MRSA virulence. Its disruption impairs membrane stability, motility, and resistance to oxidative stress, leading to attenuated virulence in a murine infection model. These findings highlight yqhG as a potential target for novel therapeutic strategies to combat MRSA infections.

## Supporting information

**S1 Table. Optical density measurements of MRSA wild-type, ΔyqhG, and complemented strains during *in vitro* growth.**
(XLSX)

**S2 Table. Survival data of BALB/c mice infected with MRSA wild-type, ΔyqhG, and complemented strains.**
(XLSX)

**S3 Table. Heart bacterial burden in mice infected with MRSA wild-type, ΔyqhG, and complemented strains.**
(XLSM)

**S4 Table. Kidney bacterial burden in mice infected with MRSA wild-type, ΔyqhG, and complemented strains.**
(XLSX)

**S5 Table. Liver bacterial burden in mice infected with MRSA wild-type, ΔyqhG, and complemented strains.**
(XLSX)

**S6 Table. Spleen bacterial burden in mice infected with MRSA wild-type, ΔyqhG, and complemented strains.**
(XLSX)

**S7 Table. Crystal violet quantification of biofilm formation in MRSA wild-type, ΔyqhG, and complemented strains.**
(XLSX)

**S8 Table. SYTOX Green fluorescence measurements for membrane permeability assessment.**
(XLSX)

**S9 Table. Colony diameter measurements from soft agar motility assays.**
(XLSX)

**S10 Table. Amount of PSMs (µg/1mL culture) in MRSA wild-type, ΔyqhG, and complemented strains.**
(XLSX)

**S11 Table. Viable CFU counts following hydrogen peroxide exposure.**
(XLSX)

## Acknowledgments

Authors have no acknowledgments to declare.

## Author contributions

**Conceptualization:** Jianhua Liao, Yuzhi Shao, Chunyan Meng.

**Data curation:** Jianhua Liao, Baoqing Liu, Yuzhi Shao, Chunyan Meng.

**Formal analysis:** Jun Cheng, Baoqing Liu.

**Funding acquisition:** Chunyan Meng.

**Investigation:** Jianhua Liao, Jun Cheng, Baoqing Liu, Yuzhi Shao.

**Methodology:** Baoqing Liu.

**Resources:** Jun Cheng.

**Software:** Jianhua Liao, Baoqing Liu, Yuzhi Shao.

**Supervision:** Jun Cheng, Yuzhi Shao.

**Validation:** Baoqing Liu, Yuzhi Shao.

**Visualization:** Jianhua Liao, Jun Cheng.

**Writing – original draft:** Jianhua Liao, Yuzhi Shao, Chunyan Meng.

**Writing – review & editing:** Jianhua Liao, Yuzhi Shao, Chunyan Meng.

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
