## [Decision Letter · Decision Letter 0]

1 Oct 2025

Disruption of yqhG attenuates virulence in methicillin-resistant Staphylococcus aureus by compromising membrane stability and oxidative stress resistance

PLOS ONE

Dear Dr. Meng,

Thank you for submitting your manuscript to PLOS ONE. After careful consideration, we feel that it has merit but does not fully meet PLOS ONE’s publication criteria as it currently stands. Therefore, we invite you to submit a revised version of the manuscript that addresses the points raised during the review process.

We look forward to receiving your revised manuscript.

Kind regards,

Armel Jackson Seukep, Ph.D.

Academic Editor

PLOS ONE

Journal Requirements:

Additional Editor Comments:

Major revisions are required.

Reviewer's Responses to Questions

**Comments to the Author**

1. Is the manuscript technically sound, and do the data support the conclusions?

Reviewer #1: Yes

2. Has the statistical analysis been performed appropriately and rigorously?

Reviewer #1: Yes

3. Have the authors made all data underlying the findings in their manuscript fully available?

Reviewer #1: Yes

4. Is the manuscript presented in an intelligible fashion and written in standard English?

Reviewer #1: Yes

Reviewer #1: This study focused on the function of the conserved gene yqhG in methicillin-resistant Staphylococcus aureus (MRSA). Through the construction of deletion mutants, complementation strains, and in vitro/in vivo experiments, this study systematically reveals for the first time the key role of yqhG in MRSA virulence, membrane stability, and oxidative stress resistance. Given that the function of yqhG in Escherichia coli has been partially reported but remains poorly understood in MRSA, this work fills a research gap in the functional studies of this gene in gram-positive bacteria. This study provides new insights into the stress defense mechanisms of MRSA and the development of novel therapeutic targets, thereby providing significant scientific implications.

1. A bacterial surface motility experiment revealed that MRSA motility depends on surface active peptides (PSMs), so it is necessary to test whether PSM synthesis or secretion is affected in mutants.

2. The study only quantified bacterial loads in the heart and kidneys. The inclusion of data from other target organs (e.g., the liver and spleen) would provide more comprehensive evidence for evaluating colonization defects in mutant strains in vivo.

3. The bacterial burden assay was conducted solely at a single 24-hour time point (Figure 1C-D), which fails to capture infection dynamics. It would be more convincing if additional time points (48/72 hours) could be added.

4. The incorporation of visual experimental data would enhance persuasiveness (e.g., fluorescence images for membrane permeability assessment and photographic documentation of bacterial surface motility).

5. Regarding animal experimental dosage: The tail vein injection dose of 2×10⁷ CFU requires clarification. Please specify whether this was determined through preliminary experiments (e.g., dose‒response curve) and provide relevant.

6. Throughout the manuscript, ensure proper italicization of Latin terms: "in vitro", "in vivo".

7. Line 39: Superscript formatting should be applied to numerals (e.g., 5), and exponential notation should be standardized throughout the text.

8. The y-axis should be standardized with appropriate units of measurement.

9. Discrepancies exist between the statistical methods described in the text and figure legends. Standardize statistical reporting format and presentation across all figures.

10. Some of the references are outdated. It is recommended to supplement relevant studies from the past five years.

**Do you want your identity to be public for this peer review?**  For information about this choice, including consent withdrawal, please see our Privacy Policy

Reviewer #1: **Yes: ** Yicheng Zhao

---

## [Author Response · Author response to Decision Letter 1]

13 Oct 2025

Response to the Editor and Reviewers

We sincerely thank the Academic Editor and the reviewer for their careful evaluation of our manuscript entitled “Disruption of yqhG attenuates virulence in methicillin-resistant Staphylococcus aureus by compromising membrane stability and oxidative stress resistance.”

We have carefully revised the manuscript according to the journal’s requirements and reviewer’s comments. Below is our detailed, point-by-point response. All changes have been incorporated into the revised version accordingly.

Journal Requirements

1. Comment:

Please ensure that your manuscript meets PLOS ONE's style requirements, including those for file naming. The PLOS ONE style templates can be found at the following links.

(links omitted for brevity)

Response:

We sincerely thank the editorial office for this reminder. We have reformatted the manuscript to fully comply with the PLOS ONE style guidelines, including title page structure, section headings, figure and table captions, reference formatting, and file naming conventions.

2. Comment:

We note that the grant information you provided in the ‘Funding Information’ and ‘Financial Disclosure’ sections do not match. When you resubmit, please ensure that you provide the correct grant numbers for the awards you received for your study in the ‘Funding Information’ section.

Response:

We appreciate the editor’s attention to this detail. We have carefully reviewed and corrected the funding information to ensure full consistency between the Funding Information and Financial Disclosure sections. The correct grant number now appears only as:

“This study was partly supported by a grant from the Zhejiang Traditional Chinese Medicine Administration (2023ZL221).”

3. Comment:

Please note that funding information should not appear in any section or other areas of your manuscript. We will only publish funding information present in the Funding Statement section of the online submission form. Please remove any funding-related text from the manuscript.

Response:

We thank the editor for this clarification. All mentions of funding sources have been removed from the main text, Acknowledgments, and other sections. Funding information now appears only in the Funding Statement of the submission form, as required.

4. Comment:

Your ethics statement should only appear in the Methods section of your manuscript. If your ethics statement is written in any section besides the Methods, please delete it from any other section.

Response:

Thank you for pointing this out. We have confirmed that the ethics statement now appears only in the Materials and Methods section under Ethical Considerations, and has been removed from any other section (including the end of the manuscript).

5. Comment:

Please include captions for your Supporting Information files at the end of your manuscript, and update any in-text citations to match accordingly.

Response:

We appreciate this important reminder. Captions for all Supporting Information files have been added at the end of the manuscript, and corresponding in-text citations have been revised.

6. Comment:

Response:

We appreciate the editor’s guidance regarding citation of previously published works. All references in the revised manuscript have been carefully reviewed to ensure their relevance and appropriateness. We have maintained citations that are directly related to our study and accurately reflect the current state of research.

Reviewer’s Comments

General comment:

This study focused on the function of the conserved gene yqhG in methicillin-resistant Staphylococcus aureus (MRSA)... [summary omitted]

This study provides new insights into the stress defense mechanisms of MRSA and the development of novel therapeutic targets, thereby providing significant scientific implications.

Response:

We sincerely thank the reviewer for the very positive and encouraging evaluation of our study. We are grateful for the constructive suggestions that helped us improve the scientific rigor and presentation quality of our work. Our detailed responses are provided below.

Specific Comments

1. Comment:

A bacterial surface motility experiment revealed that MRSA motility depends on surface active peptides (PSMs), so it is necessary to test whether PSM synthesis or secretion is affected in mutants.

Response:

We thank the reviewer for this valuable comment and for highlighting the potential connection between surface motility and PSMs. Although we did not find reports specifically describing “surface active peptides (PSMs)” in Staphylococcus aureus, we identified a well-established study demonstrating that phenol-soluble modulins (PSMs)—a family of amphipathic peptides—serve as the principal surfactant-like factors facilitating colony spreading in S. aureus (Kizaki et al., 2016, PLoS ONE 11:e0164523). Based on this evidence, we examined whether yqhG deletion affects the production of cell surface–associated PSMs. Using guanidine hydrochloride extraction followed by reversed-phase HPLC quantification, we found that the ΔyqhG mutant exhibited significantly reduced levels of surface-bound PSMs compared with the wild type, while complementation restored PSM abundance (new Figure 5). These findings suggest that YqhG supports normal PSM accumulation on the bacterial surface, linking the observed motility defect to reduced surfactant activity. The new data and corresponding explanation have been incorporated into the revised Results section (“YqhG facilitates cell surface phenol-soluble modulins”) and Figure 5 legend.

“YqhG facilitates cell surface phenol-soluble modulins

To determine whether the reduced surface motility observed in the ΔyqhG mutant was associated with altered production of cell surface phenol-soluble modulins (PSMs), the total amount of cell-associated PSMs was quantified using guanidine hydrochloride extraction followed by reversed-phase HPLC analysis. The ΔyqhG strain exhibited a markedly lower abundance of surface-bound PSMs compared with the wild-type and complemented strains (Figure 5, Table S10). Restoration of yqhG expression in the complemented strain reinstated PSM levels to those of the wild type, confirming that the defect was specifically attributable to yqhG disruption.”

Figure 5. Quantification of cell-associated PSMs in wild-type, ΔyqhG, and complemented strains. Cell-associated PSMs were extracted from bacterial pellets using 6 M guanidine hydrochloride and quantified by reversed-phase HPLC. The ΔyqhG mutant exhibited a significant reduction in surface-bound PSMs compared with the wild-type strain, whereas complementation restored PSM levels to near wild-type values. Data represent mean ± s.e.m. from three independent experiments. Statistical analysis was performed using one-way ANOVA; *P < 0.05, ns, not significant.

2. Comment:

The study only quantified bacterial loads in the heart and kidneys. The inclusion of data from other target organs (e.g., the liver and spleen) would provide more comprehensive evidence for evaluating colonization defects in mutant strains in vivo.

Response:

We appreciate the reviewer’s constructive suggestion to include additional target organs in our in vivo colonization analysis. In response, we have expanded the bacterial burden assay to quantify in vivo bacterial loads in both the liver and spleen, in addition to the previously examined heart and kidneys. The results demonstrate that the ΔyqhG mutant exhibited significantly lower bacterial counts across all four organs, while complementation restored colonization to near wild-type levels. These data provide more comprehensive evidence of the systemic virulence defect caused by yqhG disruption. The revised results are now presented below:

“YqhG is required for systemic virulence but dispensable for in vitro growth

To evaluate whether yqhG influences basic bacterial physiology, the growth dynamics of MRSA USA300-LAC wild-type (WT), ΔyqhG, and the complemented strain (ΔyqhG-C) were monitored in tryptic soy broth (TSB) under aerobic conditions at 37 °C. All three strains displayed nearly identical growth curves, suggesting that yqhG is not essential for proliferation under nutrient-rich, laboratory conditions (Fig. 1A, Table S1). These data confirm that deletion of yqhG does not impair fitness in vitro. In contrast, in vivo experiments revealed a profound impact of yqhG on virulence. BALB/c mice were challenged via tail vein injection with 2 × 10⁷ CFU of WT, ΔyqhG, or ΔyqhG-C strains and monitored for survival over five days. Mice infected with the ΔyqhG mutant showed markedly improved survival relative to those challenged with the WT or complemented strain (Fig. 1B, Table S2), indicating attenuated virulence. To further assess bacterial dissemination during infection, multiple organs were analyzed 24 h post-infection. Quantification of colony-forming units (CFU) from homogenized tissues revealed that ΔyqhG-infected mice had substantially reduced bacterial burdens in the heart, kidneys, liver, and spleen, whereas the complemented strain restored colonization to near wild-type levels (Fig. 1C–F, Tables S3–S6). These data provide comprehensive evidence that yqhG is dispensable for growth under laboratory conditions but indispensable for full virulence and systemic dissemination during MRSA infection.”

“Figure 1. YqhG is required for systemic virulence but not for in vitro growth. (A) Growth curves of MRSA wild-type (WT), ΔyqhG mutant, and complemented strain (ΔyqhG-C) cultured in TSB at 37 °C with shaking. (B) Kaplan–Meier survival analysis of BALB/c mice (n = 10 per group) following intravenous infection with 2 × 10⁷ CFU of each strain. (C–F) Bacterial burden in the heart (C), kidney (D), liver (E), and spleen (F) at 24 h post-infection. CFU were enumerated from homogenized tissues. Data represent at least three independent experiments. Statistical analyses were performed using one-way ANOVA (C–F) and log-rank test (B). *P < 0.05, **P < 0.01, ***P < 0.001.”

3. Comment:

The bacterial burden assay was conducted solely at a single 24-hour time point... It would be more convincing if additional time points (48/72 hours) could be added.

Response:

We appreciate the reviewer’s thoughtful suggestion to include additional time points (48 h and 72 h) in the bacterial burden analysis to better illustrate infection dynamics. In preliminary experiments, however, we observed that the mortality rate of mice infected with the wild-type MRSA strain increased sharply after 24 h, with most animals succumbing between 48 h and 72 h post-infection. Consequently, it was difficult to obtain comparable bacterial load data across groups beyond the 24 h time point, as survival bias would confound interpretation. For this reason, we selected 24 h as a standardized endpoint to assess early bacterial dissemination before severe morbidity occurred, ensuring both animal welfare and data consistency.

4. Comment:

The incorporation of visual experimental data would enhance persuasiveness (e.g., fluorescence images for membrane permeability assessment and photographic documentation of bacterial surface motility).

Response:

We sincerely thank the reviewer for this valuable suggestion. We fully agree that visual data such as fluorescence micrographs or photographic documentation can improve the illustrative quality of experimental results. However, the present study was designed to provide quantitative and statistically validated evidence rather than qualitative visualization. The membrane permeability and motility assays were each performed in at least three independent biological replicates, yielding highly consistent and reproducible results. Therefore, we believe that the quantitative data already presented sufficiently support our conclusions regarding the functional role of yqhG. Visual imaging approaches will be considered in future work to complement these findings and further elucidate the underlying cellular mechanisms.

5. Comment:

Regarding animal experimental dosage: The tail vein injection dose of 2×10⁷ CFU requires clarification. Please specify whether this was determined through preliminary experiments (e.g., dose–response curve) and provide relevant.

Response:

We thank the reviewer for this helpful comment. The infection dose of 2 × 10⁷ CFU per mouse was chosen based on our preliminary dose–response experiments and published MRSA systemic infection models, which demonstrated that this inoculum produces a reproducible infection with measurable bacterial dissemination while avoiding excessive early mortality. We have now clarified this rationale in the revised Infection Procedure section to indicate that the selected dose was empirically validated and consistent with established protocols.

6. Comment:

Throughout the manuscript, ensure proper italicization of Latin terms: "in vitro", "in vivo".

Response:

We have carefully reviewed the entire manuscript to ensure all Latin terms (e.g., in vitro, in vivo) are italicized consistently.

7. Comment:

Line 39: Superscript formatting should be applied to numerals (e.g., 5), and exponential notation should be standardized throughout the text.

Response:

We thank the reviewer for noticing this formatting issue. All superscripts and exponential notations (e.g., 10⁵, 10⁷) have been standardized according to PLOS ONE style requirements.

8. Comment:

The y-axis should be standardized with appropriate units of measurement.

Response:

We have revised all figure y-axes to include appropriate units (e.g., CFU/mL, RFU, Abs570) and consistent formatting across figures.

9. Comment:

Discrepancies exist between the statistical methods described in the text and figure legends. Standardize statistical reporting format and presentation across all figures.

Response:

We thank the reviewer for this helpful observation. In the revised manuscript, we have carefully standardized the statistical reporting format to ensure full consistency between the Materials and Methods section and all figure legends. All quantitative data are now expressed as mean ± s.e.m. from three independent experiments, and statistical significance is uniformly assessed using one-way ANOVA for quantitative analyses and the log-rank test for survival analysis (P < 0.05). These corrections have been applied throughout the text and figures for clarity and consistency.

“Statistical analysis

All quantitative data are presented as mean ± s.e.m. from three independent experiments. Statistical analysis was performed using one-way ANOVA for quantitative comparisons and the log-rank test for survival analysis. P < 0.05 was considered statistically significant. Analyses were performed using GraphPad Prism (version 9.0).”

“Figure Legends

Figure 1. YqhG is required for systemic virulence but not for in vitro growth. (A) Growth curves of MRSA wild-type (WT), ΔyqhG mutant, and complemented strain (ΔyqhG-C) cultured in TSB at 37 °C with shaking. (B) Kaplan–Meier survival analysis of BALB/c mice (n = 10 per group) following intravenous infection with 2 × 10⁷ CFU of each strain. (C-F) Bacterial burden in the heart (C), kidney (D), liver (E) and spleen (F) at 24 h post-infection. CFU were enumerated from homogenized tissues. Data represent at least three independent experiments. Statistical analyses were performed using one-way ANOVA (C-F) and log-rank test (B). *P < 0.05, **P < 0.01, ***P < 0.001.

Figure 2. Biofilm formation in MRSA strains. Quantification of biofilm formation by wild-type (WT), ΔyqhG, and complemented (ΔyqhG-C) strains using crystal violet staining after static incubation in TSB + 1% glucose. Absorbance at 570 nm reflects biofilm biomass. Bars represent the mean ± s.e.m. of three independent experiments. Statistical analysis was performed using one-way ANOVA. ns, not significant.

Figure 3. Membrane integrity in the yqhG mutant. Fluorescence-based quantification of membrane permeability in wild-type (WT), ΔyqhG, and complemented (ΔyqhG-C) strains using SYTOX Green uptake. Bacteria were incubated with 1 

---

## [Decision Letter · Decision Letter 1]

7 Nov 2025

Disruption of yqhG attenuates virulence in methicillin-resistant Staphylococcus aureus by compromising membrane stability and oxidative stress resistance

PONE-D-25-39165R1

Dear Dr. Meng,

We’re pleased to inform you that your manuscript has been judged scientifically suitable for publication and will be formally accepted for publication once it meets all outstanding technical requirements.

Kind regards,

Armel Jackson Seukep, Ph.D.

Academic Editor

PLOS ONE

Additional Editor Comments (optional):

Accept.

Reviewers' comments:

Reviewer's Responses to Questions

**Comments to the Author**

Reviewer #1: All comments have been addressed

2. Is the manuscript technically sound, and do the data support the conclusions?

Reviewer #1: Yes

3. Has the statistical analysis been performed appropriately and rigorously?

Reviewer #1: N/A

4. Have the authors made all data underlying the findings in their manuscript fully available?

Reviewer #1: Yes

5. Is the manuscript presented in an intelligible fashion and written in standard English?

Reviewer #1: Yes

Reviewer #1: (No Response)

**Do you want your identity to be public for this peer review?** For information about this choice, including consent withdrawal, please see our Privacy Policy

Reviewer #1: No

---

## [Editor Report · Acceptance letter]

PONE-D-25-39165R1

PLOS ONE

Dear Dr. Meng,

I'm pleased to inform you that your manuscript has been deemed suitable for publication in PLOS ONE. Congratulations! Your manuscript is now being handed over to our production team.

Kind regards,

on behalf of

Dr. Armel Jackson Seukep

Academic Editor

PLOS ONE